# End-to-end generation and evaluation of nuclei-aware histology patches using diffusion models

## Abstract

In recent years, computational pathology has witnessed remarkable progress, particularly through the adoption of deep learning techniques in segmentation and classification tasks that enhance diagnostic and prognostic workflows. Despite its importance, training effective deep learning models for these applications remains a significant challenge due to the need for large-scale annotated datasets. We present a nuclei-aware semantic tissue generation framework leveraging advancements in conditional diffusion modeling. Our framework generates high-quality synthetic tissue patches that are inherently annotated with instances of six distinct nuclei types. We demonstrate the efficacy of generated samples through extensive quantitative and expert evaluation.

## 1 Introduction

Histopathology relies on the microscopic examination of hematoxylin and eosin (H&E) stained tissue biopsies to identify visual evidence of diseases, including various types of cancer. Accurate diagnosis often hinges on the expertise and prior experience of pathologists, particularly their exposure to a wide range of disease subtypes. However, rare disease variants pose challenges due to their limited representation in learning datasets, making visual identification difficult.

In recent years, deep learning methods have sought to address these challenges by developing accurate probabilistic models to assist in diagnosis. Segmentation models, for instance, have been widely applied to localize and classify different nuclei types in tissue samples.

Generative models which synthesize realistic histologic patches with specific features, including patterns associated with rare disease subtypes, enable the creation of unlimited annotated datasets for training both humans and deep learning models. These curated datasets can mitigate biases, improve generalization, and ensure equal representation of disease subtypes. Furthermore, synthetic datasets provide a solution to privacy concerns surrounding medical data sharing while also reducing the time, labor, and other costs associated with annotating large medical datasets.

Recent advancements in denoising diffusion probabilistic models (DDPMs) (Ho et al., 2020) have demonstrated remarkable success in generating high-quality real-world images, both conditionally and unconditionally. Notably, the semantic diffusion model (SDM) (Wang et al., 2022) has shown the capability to generate images from semantic layouts, hinting the potential of DDPMs in medical image synthesis.

In this work,

- We design a first-of-its-kind conditional diffusion model tailored for histology patch synthesis.

- We utilize the Lizard dataset (Graham et al., 2021), a comprehensive collection of colon histology images, achieving state-of-the-art results in synthetic tissue generation.

- We conduct extensive qualitative, quantitative, and ablative analyses to demonstrate the efficacy of our framework and its potential applications in pathology pedagogy, model validation, and dataset augmentation.

By enabling pixel-perfect localization and diversity in synthetic tissue generation, our approach addresses key limitations in existing datasets and opens new avenues for both computational and clinical pathology.

## 2 Background

Deep learning based generative models for histopathology images have seen tremendous progress in recent years due to advances in digital pathology, compute power, and neural network architectures. Several GAN-based generative models have been proposed to generate histology patches (Levine et al., 2020; Xue et al., 2021; Zhou & Yin, 2022). However, GANs suffer from problems of frequent mode collapse and overfitting their discriminator (Xiao et al., 2021). It is also challenging to capture long-tailed distributions and synthesize rare samples from imbalanced datasets using GANs. More recently, denoising diffusion models have been shown to generate highly compelling images by incrementally adding information to noise (Ho et al., 2020). Success of diffusion models in generating realistic images led to various conditional (Kawar et al., 2022; Saharia et al., 2022a;b) and unconditional (Dhariwal & Nichol, 2021; Ho et al., 2022; Nichol & Dhariwal, 2021) diffusion models that generate realistic samples with high fidelity. Following this, a morphology-focused diffusion model has been presented for generating tissue patches based on genotype (Moghadam et al., 2023). Semantic image synthesis is a task involving generating diverse realistic images from semantic layouts. GAN-based semantic image synthesis works (Tan et al., 2021a;b; Park et al., 2019) generally struggled at generating high quality and enforcing semantic correspondence at the same time. To this end, a semantic diffusion model has been proposed that uses conditional denoising diffusion probabilistic model and achieves both better fidelity and diversity (Wang et al., 2022). We use this progress in the field of conditional diffusion models and semantic image synthesis to formulate our framework.

## 3 Nuclei Aware Diffusion Model

A conditional diffusion model aims to maximize the likelihood $p_\theta(x_0 \mid y)$, where data $x_0 \sim q(x_0 \mid y)$, $y$ being the conditioning signal. The forward process of a diffusion model consists of systematically destroys the information in a given sample using a markovian chain of Gaussian noise addition steps. And the reverse diffusion process incrementally adds information by denoising a corrupted sample. In conditional diffusion models the forward diffusion process can ignore the conditioning signal and Gaussian noise can be incrementally added to corrupt the data sample $x_0$ using the same process as described in DDPM Nichol & Dhariwal (2021). However, the denoising process is modified to incorporate the conditioning signal and is defined as a Markov chain with learned Gaussian transitions starting from pure noise, $p(x_T) \sim \mathcal{N}(0, \mathbf{I})$ and is parameterized as a neural network with parameters $\theta$,

$$p_\theta(x_{0:T} \mid y) = p(x_T) \prod_{t=1}^{T} p_\theta(x_{t-1} \mid x_t, y). \tag{1}$$

Hence, for each denoising step from $t$ to $t-1$,

$$p_\theta(x_{t-1} \mid x_t, y) = \mathcal{N}(x_{t-1}; \mu_\theta(x_t, y, t), \Sigma_\theta(x_t, y, t)). \tag{2}$$

The parameters $\theta$ are optimized using gradient descent. During optimization, time step $t$ is sampled uniformly, and the expectation $E_{t,x_0,y,\epsilon}$ is used to estimate the loss. The denoising neural network can be parameterized in various ways. In our work, a noise prediction-based formulation results in superior image quality. Consequently, the denoising model is trained to predict the noise added to the input image given the semantic layout $y$ and the time step $t$ using the loss described below:

$$L_{\text{simple}} = E_{t,x,\epsilon} \left[ \| \epsilon - \epsilon_\theta(x_t, y, t) \|_2 \right]. \tag{3}$$

It is important to note that the given simplified loss function does not provide a training signal for $\Sigma_\theta(x_t, y, t)$. To address this, following the improved DDPMs strategy (Watson et al., 2021), a network is trained to predict an interpolation coefficient $v$ for each dimension. This coefficient is then converted into variances,

$$\Sigma_\theta(x_t, y, t) = \exp\left( v \log \beta_t + (1-v) \log \widetilde{\beta}_t \right). \tag{4}$$

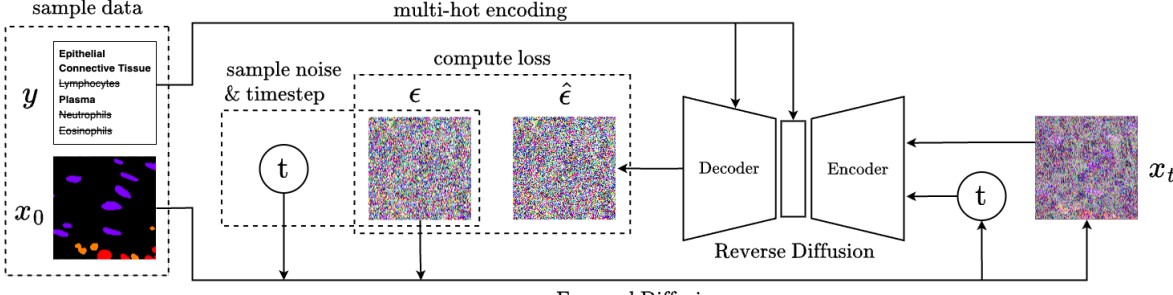

Figure 1: **Training Nuclei Mask Generation Model:** Sample a real nuclei mask $x_0$ and corresponding multi-hot encoded nuclei types $y$ from the dataset. Sample timestep $t$ and Gaussian noise $\epsilon$ to perform forward diffusion and generate noised input $x_t$. The corrupted image $x_t$, timestep $t$, and nuclei types condition $y$ are then fed into the denoising model which predicts $\hat{\epsilon}$ as the amount of noise added to the model. Original noise $\epsilon$ and prediction $\hat{\epsilon}$ are used to compute the loss in equation 3.

This is then directly optimized using $L_{vlb}$, which is,

$$L_{vlb} = D_{KL}(p_\theta(x_{t-1} \mid x_t, y) \parallel q(x_{t-1} \mid x_t, x_0)) \tag{5}$$

During this optimization, a stop gradient is applied to $\epsilon(x_t, y, t)$, allowing overall $L_{vlb}$ to guide $\Sigma_\theta(x_t, y, t)$, while $L_{\text{simple}}$ in equation 3 primarily guides $\epsilon(x_t, y, t)$. The overall loss is then a weighted sum of these two objectives, as follows:

$$L_{\text{hybrid}} = L_{\text{simple}} + \lambda L_{\text{vlb}}. \tag{6}$$

### 3.1 Generating Nuclei Mask

We use a diffusion model conditioned on the presence of six distinct cell types—epithelial cells, lymphocytes, connective tissue cells, neutrophils, plasma cells, and eosinophils. A multi-hot encoded vector, $y_{\text{Ntypes}}$, specifying the nuclei types to include in the mask is used to control the denoising model approximating the reverse diffusion process. The model is trained on the annotation masks in the lizard dataset. The dataset contains histologic patches where each nucleus is pixel-level labeled according to its type. During training, we add a linear transformation of the multi-hot vector to the time-embedding of the U-Net backbone as $t_{\text{updated}} = t + f_\theta(y_{\text{Ntypes}})$.

During training, the model learns to reverse the diffusion process in a manner conditioned on the specified cell types, effectively generating semantic masks that includes only the cell indicated by the multi-hot vector. This approach enables flexible and controllable generation of histological nuclei masks. These synthetic semantic masks can then be used with the trained model that is described in the following sections to enable infinite histological patch generation that is already annotated.

### 3.2 Semantic Mask Conditioned Patch Generation

To enhance our neural network noise predictor $\epsilon_\theta(x_t, y, t)$'s ability to process nuclei semantic map information, we adapt a U-Net architecture inspired by Wang et al. (Wang et al., 2022). This modified architecture innovatively integrates time step and semantic information through strategic feature injection.

**Encoder:** The encoder processes the noisy image using semantic diffusion encoder residual blocks and attention blocks. These residual blocks incorporate convolution, SiLU activation, and group normalization. The SiLU activation function (Paul et al., 2022), defined as $f(x) = x \cdot \text{sigmoid}(x)$, demonstrates superior performance compared to ReLU in deeper models.

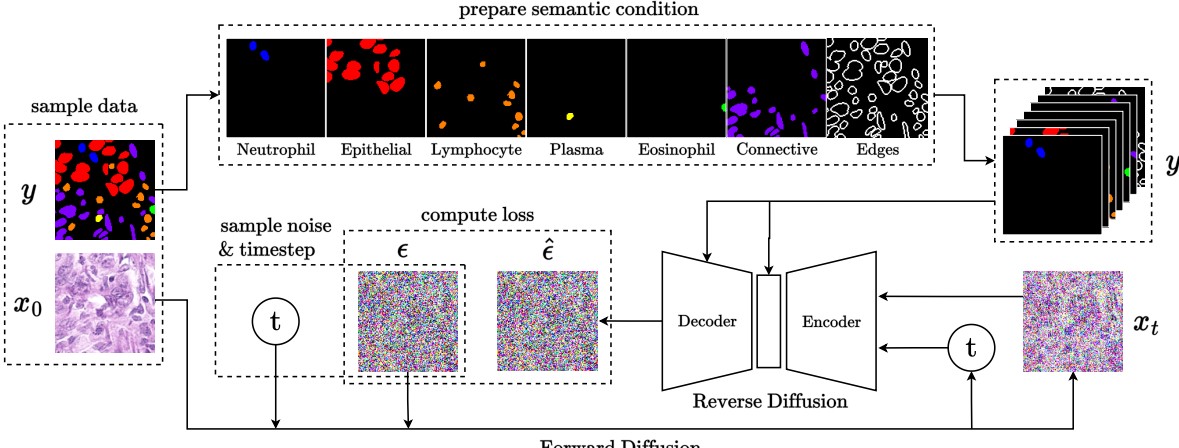

Figure 2: **Training Patch Generation Model:** Given a real image $x_0$ and semantic mask $y$, we construct the conditioning signal by expanding the mask and adding an instance edge map. We sample timestep $t$ and noise $\epsilon$ to perform forward diffusion and generate the noised input $x_t$. The corrupted image $x_t$, timestep $t$, and semantic condition $y$ are then fed into the denoising model which predicts $\hat{\epsilon}$ as the amount of noise added to the model. Original noise $\epsilon$ and prediction $\hat{\epsilon}$ are used to compute the loss in equation 3.

Time step injection occurs through a learnable scaling and shifting mechanism of intermediate activations. Specifically, for each feature transformation, we use learnable weight $w(t) \in \mathbb{R}$ and bias $b(t) \in \mathbb{R}$, mathematically expressed as $f_{i+1} = w(t) \cdot f_i + b(t)$, where $f_i, f_{i+1} \in \mathbb{R}$ represent input and output features.

**Decoder:** The semantic label map $y_{\text{Nmask}}$ is integrated into the decoder's denoising network through semantic diffusion decoder residual blocks using multi-layer spatially adaptive normalization. Unlike the encoder's group normalization, this approach employs a spatially-adaptive normalization technique.

The feature transformation is mathematically represented as:

$$f^{i+1} = \gamma^i(y_{\text{Nmask}}) \cdot \text{Norm}\left(f^i\right) + \beta^i(y_{\text{Nmask}}), \tag{7}$$

where $f^i$ and $f^{i+1}$ are input and output features, $\text{Norm}(\cdot)$ denotes parameter-free group normalization, and $\gamma^i(y_{\text{Nmask}}), \beta^i(y_{\text{Nmask}})$ represent spatially-adaptive learned weights and biases derived from the semantic layout. The conditioning signal is constructed using the semantic mask, with each channel corresponding to a unique nuclei type. Additionally, a mask containing the edges of all nuclei is concatenated to further distinguish nuclei instances, enhancing the network's spatial understanding.

### 3.3 Data Description

We use the publicly available Lizard dataset Graham et al. (2021), comprising histology image regions of colon tissue from six distinct sites. These tissue images, obtained at a $20\times$ objective magnification, are annotated for epithelial cells, connective tissue cells, lymphocytes, plasma cells, neutrophils, and eosinophils. The dataset consists of 238 tissue images, with an average size of $1055 \times 934$ pixels. For computational viability, all the tissue images were divided into smaller image patches of $128 \times 128$ pixels at $20\times$ objective magnification. Patching is performed with a 50% overlap in neighboring patches to ensure the information at the patch boundary is not lost. Patches with less than 50% tissue area were excluded from consideration. The tissue images yield a total of $59,726$ patches.

| Method | Tissue type | Conditioning | FID($\downarrow$) | IS($\uparrow$) |
|--------|-------------|--------------|---------|--------|
| BigGAN (Brock et al., 2018) | bladder | none | 158.4 | - |
| AttributeGAN (Ye et al., 2021) | bladder | attributes | 53.6 | - |
| ProGAN (Karras et al., 2017) | glioma | morphology | 53.8 | 1.7 |
| Morph-Diffusion (Moghadam et al., 2023) | glioma | morphology | 20.1 | 2.1 |
| Morph-Diffusion[*] (Moghadam et al., 2023) | colon | morphology | 18.8 | 2.2 |
| NASDM (Real Masks) | colon | nuclei mask | **14.1** | **2.7** |
| NASDM++ (Generated Masks) | colon | syn. nuclei mask | **15.2** | **2.6** |

Table 1: **Quantitative Assessment:** We report the performance of our method using Fréchet Inception Distance (FID) and Inception Score (IS) with the metrics reported in existing works. (-) denotes that corresponding information was not reported in original work. [*]Note that performance reported for best competing method on the colon data is from our own implementation, performances for both this and our method should improve with better tuning. Please refer to our github repo for updated statistics.

## 4 Experiments

In this section, we first describe our implementation details and training procedure. We then perform quantitative and qualitative assessments to demonstrate the efficacy of our nuclei-aware semantic histopathology generation model.

### 4.1 Implementation Details

We implement our diffusion models for patch and mask generation using a semantic UNet architecture (as detailed in Section 3.2), training the model with the objective specified in equation equation 6. Drawing inspiration from previous research by Nichol et al. (Nichol & Dhariwal, 2021), we set the trade-off parameter $\lambda$ at 0.001. For model optimization, we employ the AdamW optimizer and incorporate an exponential moving average (EMA) of the denoising network weights, applying a decay rate of 0.999. Following the methodology established in DDPM (Ho et al., 2020), our implementation uses a total of 1000 diffusion steps with a linear noising schedule corresponding to the timestep $t$ for the forward process. Our training procedure involves an initial phase with a learning rate of $1e - 4$, followed by a learning rate reduction to $2e - 5$ to facilitate fine-tuning. We introduce a dropout rate of 0.2 to augment the classifier-free guidance capabilities during the sampling phase. The entire framework is developed using PyTorch and trained across 4 NVIDIA Tesla A100 GPUs, with each GPU processing a batch size of 40. Upon publication or upon request, we will make the source code available.

### 4.2 Generative Metrics Evaluation

To the best of our knowledge, ours is the only work that is able to synthesize histology images given a semantic mask, making a direct quantitative comparison tricky. However, the standard generative metric Fréchet Inception Distance (FID) measures the distance between distributions of generated and real images in the Inception-V3 (Kynkäänniemi et al., 2022) latent space, where a lower FID indicates that the model is able to generate images that are very similar to real data. Therefore, we compare FID and IS metrics with the values reported in existing works (Ye et al., 2021; Moghadam et al., 2023) (ref. Table 1) in their own settings. We can observe that our method outperforms all existing methods including both GANs-based methods as well as the recently proposed morphology-focused generative diffusion model.

### 4.3 Downstream Task Evaluation

In this study, our objective is twofold: (1) to generate synthetic tissue patches from annotated semantic masks using a nuclei-aware semantic diffusion model and (2) to train and evaluate nuclei segmentation models investigating the potential of synthetic data in enhancing downstream segmentation performance. In

this section, we (1) describe the datasets used for training and validating the patch generation and HoVerNet models, along with the steps for data preparation, (2) detail the process of nuclei semantics conditioned patch generation using our method, and (3) outline the training and evaluation of nuclei segmentation model for downstream experiments.

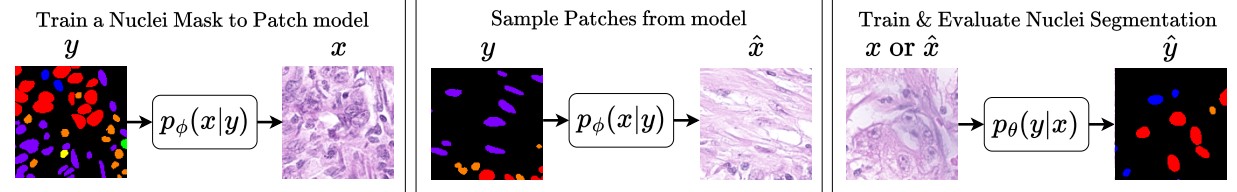

Figure 3: **Overall approach:** We have patches $x$ sampled from conditional data distribution, $x \sim q(x \mid y)$, and masks $y$ as the conditioning signal. We train a conditional generative model $p_\phi(x \mid y)$ (left), sample synthetic images (middle), and then evaluate the efficacy of synthetic images in training nuclei segmentation models $p_\theta(y \mid x)$ (right). Here $\phi$ and $\theta$ represent the parameters of patch generation and HoVerNet models.

In order to investigate the effectiveness of synthetic datasets in improving nuclei segmentation models, we perform three evaluation experiments: (1) Addition of Synthetic Patches (4.3.3) to the training of nuclei segmentation models. This experiment explores the impact of supplementing the training dataset of the HoVerNet model by adding synthetic images generated from our generative model. (2) Replacement with Synthetic Patches (4.3.4) for nuclei segmentation training. In this experiment, we evaluate the performance of segmentation models trained on datasets comprising different ratios of real to synthetic patches. And (3) Synthetic vs Real Patches (4.4) for training downstream nuclei segmentation models. This experiment determines how effective synthetically generated patches are for training nuclei segmentation models compared to their real counterparts. In all following experiments, the dice score is reported on a held-out real test set described in Table 2. The models are validated after every two epochs during training, and we report the metrics of the best-performing model on the test set.

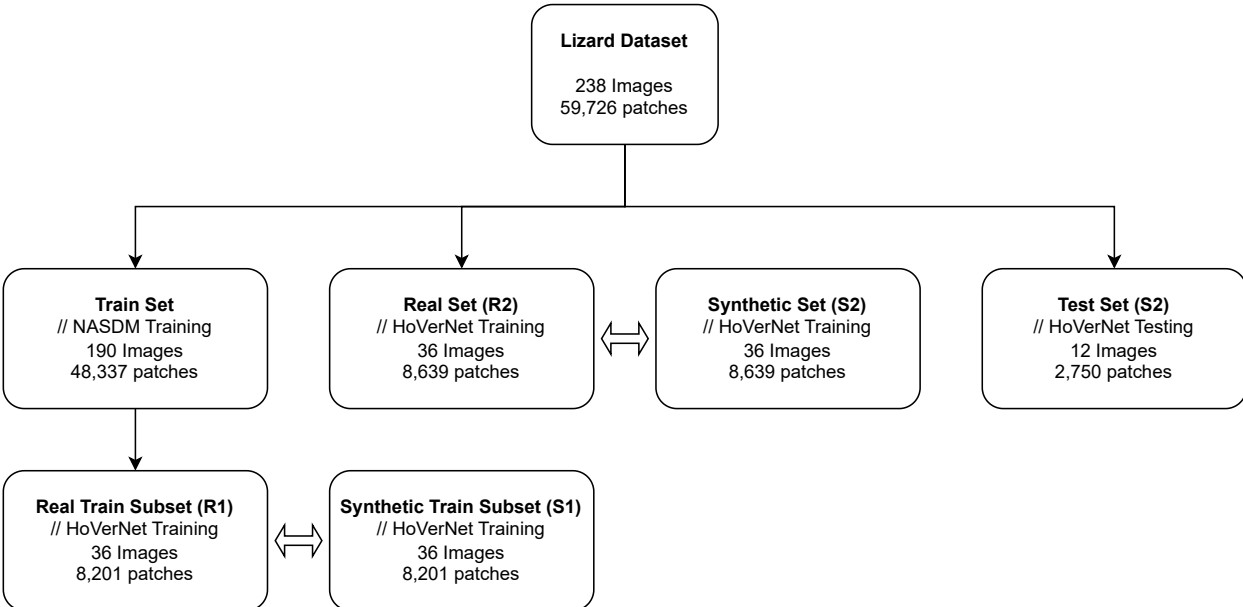

Figure 4: **Overview of data:** The figure describes the different subsets of Lizard dataset used for training and evaluation of the patch generation and HoVerNet models in our experiments. Refer Table 2 for further details.

| Dataset | ID | Type | NASDM Training | HoVerNet Training | # Images | # Patches |
|---------|-----|------|----------------|-------------------|----------|-----------|
| Train Set | Train | Real | ✓ | ✗ | 190 | 48,337 |
| Real Train Subset | R1 | Real | ✓ | ✓ | 36 | 8,201 |
| Synthetic Train Subset | S1 | Synthetic | ✗ | ✓ | 36 | 8,201 |
| Real Set | R2 | Real | ✗ | ✓ | 36 | 8,639 |
| Synthetic Set | S2 | Synthetic | ✗ | ✓ | 36 | 8,639 |
| Test Set | Test | Real | ✗ | ✗ | 12 | 2,750 |

Table 2: **Overview of data:** Different subsets of Lizard dataset used for training and evaluation of the patch generation and HoVerNet models.

### 4.3.1 Dataset Setup

We train the generative model on a Train Set containing 190 tissue images from the Lizard dataset tiled into $48,337$ patches. From this Train Set we select a subset R1 consisting of 36 images ($8,201$ patches) and generate a corresponding synthetic subset S1 from NASDM using R1's real nuclei masks. From the Lizard dataset, we select another subset R2 comprising of 36 images tiled into $8,639$ smaller patches. The images in R2 are not a part of the Train Set used for training the NASDM model. We also generate a synthetic set S2 using the nuclei masks of R2. Lastly, we reserve 12 images with $2,750$ patches, not included in any of the sets mentioned above, for testing the segmentation models trained in downstream experiments. Table 2 provides the details of the subsets of the dataset used in different tasks along with their designated nomenclature, number of images, and number of patches.

### 4.3.2 Nuclei Segmentation Model Setup

In all our downstream experiments, we employ HoVerNet for nuclear segmentation. The training of HoVerNet is a two-stage process. In the initial stage, the model is initialized with pre-trained weights from the ImageNet dataset, and the decoder is trained exclusively for 50 epochs with a batch size of 16. In the second stage, all the layers are fine-tuned for another 50 epochs. In both stages, we train the model using Adam optimizer with an initial learning rate of $10^{-4}$ and then reduce it to a rate of $10^{-5}$ after 25 epochs. We use the best-performing model over the hundred epochs for testing. To assess the model's performance we compute the Dice score and Mean Intersection over Union (IoU) of the predicted masks with respect to their actual counterparts. The final metric is obtained by averaging scores first by channel and then by batch, providing a comprehensive evaluation of performance.

### 4.3.3 Addition of Synthetic Patches

In this experiment, our objective is to assess the effectiveness of using synthetically generated data to augment existing datasets for nuclei segmentation tasks. Initially, we train a segmentation model exclusively on a 25% subset of R2. We then accumulate synthetic images from S2, which correspond to the masks of the remaining 75% of R2. We progressively incorporate subsets of these images from the synthetic image set S2 into the training. Note that the size of the training dataset increases with the addition of additional images. Also, note that we do not use all the images in the set R2 as the base set to make sure that the added images correspond to new masks that do not exist in the base training set we start with. The Dice scores and mean IoU of all the models on the Test set are presented in Table 3. We observe that as the training data is supplemented with synthetic patches, there is a discernible improvement in the model performance. This trend of gradual improvement underscores the beneficial impact of synthetically generated images in augmenting datasets and ultimately enhancing the accuracy of segmentation tasks.

Observations from this experiment support the contention that synthetic images generated using a state-of-the-art conditional diffusion model are already useful for augmenting existing expertly annotated datasets to improve the performances of downstream nuclei segmentation models trained on them.

| Data | # Patches | Dice Score | Mean IoU |
|---|---|---|---|
| 25% R2 | 2,159 | $0.7713 \pm 0.0005$ | $0.6409 \pm 0.0008$ |
| 25% R2 + 25% S2 | 4,318 | $0.7869 \pm 0.0007$ | $0.6608 \pm 0.0012$ |
| 25% R2 + 50% S2 | 6,478 | $0.7993 \pm 0.0006$ | $0.6771 \pm 0.0005$ |
| 25% R2 + 75% S2 | 8,639 | $\mathbf{0.8092 \pm 0.0004}$ | $\mathbf{0.6889 \pm 0.0007}$ |

Table 3: **Addition of Synthetic Patches:** Segmentation performance of the nuclei segmentation model with consecutive augmenting of training set using synthetic data. We report the mean and standard deviation across three runs for both metrics.

| Data | Dice Score | Mean IoU |
|---|---|---|
| R2 | $0.8091 \pm 0.0012$ | $0.6890 \pm 0.0014$ |
| 75% R2 + 25% S2 | $\mathbf{0.8098 \pm 0.0007}$ | $\mathbf{0.6902 \pm 0.0009}$ |
| 50% R2 + 50% S2 | $0.8097 \pm 0.0006$ | $0.6898 \pm 0.0007$ |
| 25% R2 + 75% S2 | $0.8092 \pm 0.0004$ | $0.6889 \pm 0.0007$ |
| S2 | $0.8087 \pm 0.0004$ | $0.6886 \pm 0.0002$ |

Table 4: **Replacement with Synthetic Patches:** Performance of models trained on real data, synthetic data, and different combinations of both, given the same set of annotation masks. We report the mean and standard deviation across three runs for both metrics.

### 4.3.4 Replacement with Synthetic Patches

In this experiment, our goal is to compare the performance of a nuclei segmentation model when trained entirely on real data against when trained solely on synthetic data. For further clarity, we also evaluate models trained with combinations of real and synthetic data in different ratios while keeping the size of the dataset constant. The training utilized set R2 for real images and set S2 for synthetic images. Note that the set S2 is generated using the masks in the set R2. We first train a segmentation model on only R2 and then systematically replace a portion of patches in R2 with corresponding synthetic patches from S2, ensuring the total number of images and the masks used stay the same. The Dice score and mean IoU on the Test Set are detailed in Table 4. Notably, performance across all runs is comparable indicating that there is no loss of performance on replacement with synthetic patches. This finding indicates that synthetic data performs just as effectively, if not better, in the training of nuclei segmentation problems.

### 4.4 Synthetic vs Manual Annotations

In this experiment, our objective is to assess whether synthetic patches generated using the same masks used for training the generative model yield a better and more precise set of annotations than the real patches themselves. This experiment tests the intuition that the generative model should be able to correct for manual errors between annotators and generate synthetic patches that are more consistent with the masks than their real counterparts. To test this hypothesis, we strategically select a subset of the training set used to train the generative model, denoted as R1, and generate synthetic patches based on their corresponding annotations, forming set S1. We employed both R1 and S1 to train the HoVerNet model independently. The rationale behind this approach was to evaluate whether the more precise annotations derived from the generative model could result in a more accurate representation of nuclei boundaries, thereby potentially yielding a superior Dice score or mean IoU. The results are reported in Table 5. The comparative analysis of performance in both cases revealed notable consistency. However, it is crucial to acknowledge that the models are evaluated using manually annotated patches in the test set.

### 4.5 Expert Evaluation

We have two board-certified pathologists review the synthetic patches generated using both real and synthetic masks as the condition. We use 60 patches for this review, 20 from the real set with their corresponding

| Data | Dice Score | Mean IoU |
|------|------------|----------|
| R1 | **0.8065 ± 0.0005** | **0.6854 ± 0.0007** |
| S1 | 0.8053 ± 0.0003 | 0.6840 ± 0.0003 |

Table 5: **Synthetic vs Manual Annotations:** This table showcases the results of our investigation into the efficacy of annotations of synthetic patches generated by the generative model. We report the mean and standard deviation across three runs for both metrics.

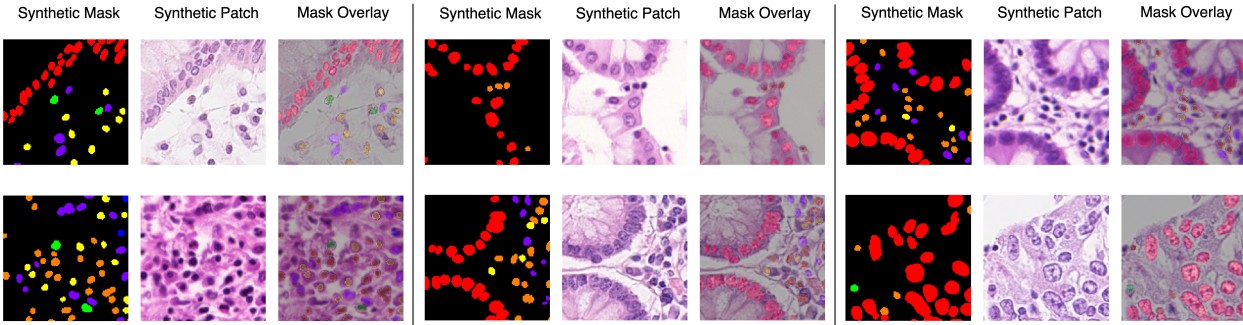

Figure 5: **Generation using synthetic masks:** We generate synthetic masks in different nuclei environments and these use these patches to generate synthetic tissue patches to demonstrate the proficiency of the model to generate realistic nuclei arrangements. Red: Epithelial cells, Purple: Connective tissue cells, Orange: Lymphocytes, Yellow: Plasma cells, Purple: Neutrophils, and Green: Eosinophils.

masks, 20 synthetic patches generated using real masks, 20 synthetic patches generated using synthetic masks from our mask generation model. The evaluation is performed on four criteria. The experts evaluate (1) whether the boundaries of the nuclei in synthetic patches match the boundaries of the nuclei in the mask; (2) if each nuclei type in the mask is accurately labeled in the evaluated patch; (3) if the model has failed to generate a nuclei present in the mask; (4) if there are any excess nuclei in the final patch that are not present in the mask. Our aim is that by evaluating these four criteria we capture the degree of realism in the patches generated from synthetic masks. A copy of the survey used for the review can be found on a public typeform[1].

The evaluation results (Figure. 6) indicate that the synthetic patches generated using both real and synthetic masks perform comparably across all four criteria. The board-certified pathologists found no significant differences in the accuracy of boundary alignment, nuclei type labeling, omission errors, or excess nuclei presence between the two types of synthetic patches. This suggests that our mask generation model produces synthetic masks that are effective in guiding high-fidelity patch synthesis, achieving a level of realism similar to that of patches generated using real masks.

## 5 Limitations

**Evaluation based on patches**: In this study, experts evaluated images within a controlled and specific setting, analyzing $128 \times 128$ pixel patches of Whole Slide Images (WSI). This approach deviates from the typical histopathologic evaluation wherein pathologists examine tissue holistically at varying levels of magnification. The focus on small image patches may limit the evaluators' ability to incorporate broader context of the tissue architecture, thus influencing assessment accuracy. Therefore, our evaluation should be interpreted within the confines of this artificial setup, and caution must be exercised when extending these results to more conventional histologic evaluations.

---

[1] https://l7d0z1f5um1.typeform.com/to/IkAbnEOv

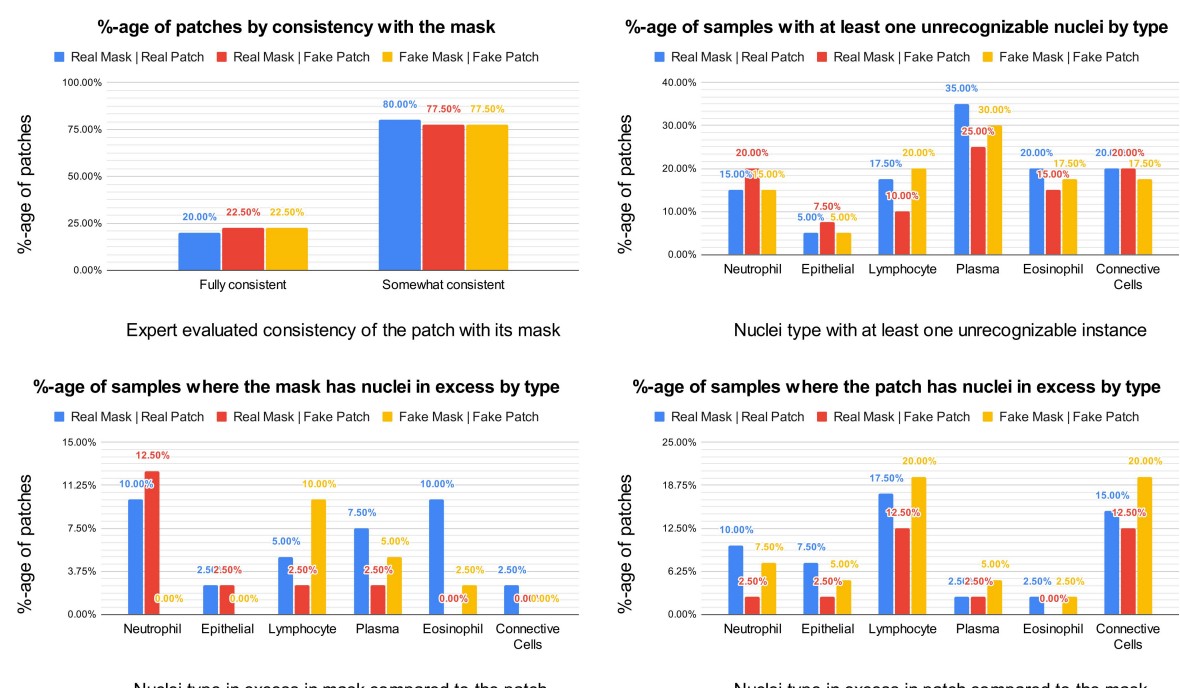

Figure 6: **Qualitative Review:** Compiled results from pathologist review. We have experts assess patches for, **top-left:** consistency of the patch with the corresponding mask, **top-right:** instances of unrecognizable nuclei types with respect to the annotated mask for each type in the patch, **bottom-left:** excess instances of nuclei in the mask with respect to the patch for each type, and **bottom-right:** excess instances of nuclei in the patch with respect to the mask for each type.

**Scope of the evaluation**: Moreover, the evaluation criteria were restricted to the four dimensions explicitly mentioned previously with the goal of assessing model performance. They do not encompass the complexity of histopathologic diagnosis, which includes nuanced morphological patterns and clinical context. As a result, this limited scope may not fully capture the model's ability to generalize beyond these criteria, nor does it account for all possible ways in which the generated patches might be useful or flawed in broader medical practice. We recognize that there may be medically relevant patterns which the model does not replicate with high fidelity. These patterns may escape detection in the qualitative assessments made by our small panel of experts.

**Size of the expert panel**: Notably, our panel consists of only two board-certified pathologists, which may introduce an element of subjectivity in the evaluation. A larger, more diverse panel comprised of attending pathologists and Pathology residents would provide a more robust assessment of our model's outputs, accounting for differences in evaluator experience and comfort working with model outputs.

**Generalization to other tissue types**: Furthermore, the model was specifically demonstrated on colonic tissue. It remains uncertain if the model will have similar performance when applied to other tissues or disease contexts. Histologic structures and disease presentations vary between tissue types, and thus, the model's ability to generalize beyond the colon cannot not be assumed. Future research should aim to extend the evaluation to include other tissue and disease states to gauge generalizability and performance across clinical scenarios.

**Biases in the generated samples**: One relevant limitation inherent to generative models is that they can only generate samples resembling the training data. As such, the model is inherently biased toward replicating familiar patterns, and it may not generate plausible histologic representations that are unrepresented in

the training set. This replication of bias is a well-known issue with generative models, as they do not possess an intrinsic mechanism to correct for biases present in the training data. Consequently, while our model demonstrates a reasonable ability to capture and replicate the training data distribution, it must be recognized that any biases or limitations in the training data will be reflected in the generated samples.

## 6 Future Work

**Expanding conditional signals**: In future work, it will be valuable to explore additional conditioning mechanisms that could improve the model's ability to generate more diverse and contextually accurate patches. For instance, conditioning the patch generation process on properties such as stain-distribution, tissue-type, disease-type, and other relevant clinical variables could allow the model to better capture the specific characteristics of different histological settings in real-world medical practice. This would also enable the model to adapt to the unique characteristics of different staining protocols, which vary between labs and can affect the appearance of histological samples.

**Generating larger tissue areas**: Furthermore, an interesting avenue for future research would be to explore the generation of patches conditioned on neighboring patches. This approach could enable the generation of larger tissue regions by stitching together individual patches, thus allowing for a more holistic representation of tissue architecture. By considering the spatial relationships between neighboring patches, the model could capture larger-scale tissue patterns that are critical for accurate histological analysis. This could open up new possibilities for the application of generative models in histopathology, enabling the synthesis of realistic tissue sections that can be used for various research and diagnostic purposes.

**Addressing biases in future work**: Finally, future studies could also consider further refining the model to address the biases present in the training data. Techniques such as domain adaptation, adversarial training, or incorporating real-world clinical feedback could be explored to mitigate these biases and ensure that the model produces more representative and equitable outputs across different tissue types and disease conditions.

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
