# OpenReview forum: "End-to-end generation and evaluation of nuclei-aware histology patches using diffusion models"
_TMLR — Rejected by TMLR_

### Review · Reviewer_nSZK · 2025-04-27

**Summary Of Contributions:**

The paper addresses a problem in computational pathology: the generation of annotated histology patches using conditional diffusion models. The idea of nuclei-aware conditioning is timely and the proposed two-stage diffusion model for generating both nuclei masks and corresponding tissue patches is methodologically sound. The results are reasonable, especially the evidence showing that models trained on synthetic patches can perform comparably to those trained on real data. The experiments are thoughtfully structured, and the writing is clear.

**Audience:**

Yes

**Broader Impact Concerns:**

The model is trained on a single dataset without substantial discussion or experimental analysis of potential biases. There is a risk that any dataset-specific artifacts, sampling biases, or class imbalances could be propagated and amplified in the synthetic data. This could inadvertently impact downstream clinical models trained on generated data, reinforcing existing biases.

The submission would benefit from an explicit Broader Impact Statement discussing the ethical considerations of synthetic data use in healthcare, risks of bias propagation, and the importance of careful deployment and validation strategies.

**Claims And Evidence:**

Yes

**Requested Changes:**

- The experiments are currently limited to the Lizard dataset (colon histology). To support claims of generalizability, the authors should evaluate their framework on additional datasets such as PanNuke, CoNSeP, or CAMELYON. This would substantiate the applicability of the approach across tissue types and disease variations.
- Important recent work on histopathology image synthesis, such as URCDM [1] and selective synthetic augmentation [2], should be compared against. At a minimum, a thorough discussion of these approaches and their relation to the proposed method should be included.
- Beyond segmentation with HoVerNet, assess the utility of the generated data for other clinically relevant tasks (e.g., disease classification, tumour grading) to demonstrate broader impact.
- Extend the expert evaluation to involve more pathologists and include assessments of larger tissue regions (not only small patches) to better reflect realistic diagnostic tasks.
- Analyse potential biases inherited from the Lizard dataset and explore how they might propagate or be mitigated during synthesis. Simple experiments, such as examining class balance or feature diversity across synthetic samples, would be valuable.
- More explicitly describe and highlight how the proposed model differs architecturally and algorithmically from baseline semantic diffusion models such as Wang et al. (2022) [6].
- Broaden the related work section to cover more recent papers on conditional generation in histopathology, e.g., Patch-to-WHO models or morphology-preserving diffusion strategies.

**Strengths And Weaknesses:**

strengths:
- The two-stage framework (nuclei mask generation followed by tissue patch generation) is methodologically sound and tailored for nuclei-aware histology synthesis. The nuclei conditioning improves control and interpretability of generated samples.
- The authors perform extensive experiments including standard generative metrics (FID, IS), downstream task evaluation (nuclei segmentation with HoVerNet), and expert-based qualitative assessment with pathologists.
- The paper is well organized, making it relatively easy to follow, with each component of the pipeline and experiment clearly explained.


weaknesses:
- The paper heavily leans on existing diffusion-based generation methods with only relatively incremental innovation, adapting semantic diffusion to nuclei-aware histopathology data. While the application is valuable, it lacks a fundamentally new conceptual contribution in the modelling side beyond domain adaptation. In particular, the overall architecture and loss formulations are very close to existing work in semantic diffusion (e.g., Wang et al. 2022 [6]) without sufficient differentiation or novel architectural or algorithmic insights.

- The evaluation, while extensive, is limited to a single dataset (Lizard) and a narrow application domain (colon histology). The claims regarding generalizability remain speculative and are not experimentally supported. Broader evaluations, for example across different tissues or disease types, would be expected for a higher-tier venue like TMLR.

- While FID and IS are reported, these metrics have known limitations when evaluating structured biomedical images. Clinically relevant, diagnostic downstream tasks beyond segmentation should be evaluated to provide a better picture about the impact of the method.
Furthermore, comparisons are limited mostly to GAN-based models and a morphology-focused diffusion model. Direct comparison to more domain-specific prior work such as [1-5] and other recent methods targeting histopathology synthesis, or selective synthetic augmentation approaches [2] are not discussed or compared against.

- The expert evaluation is a nice touch but limited by the extremely small panel (only two pathologists) and constrained context (small patches rather than larger histological context).

- As the generation is heavily conditioned on a specific dataset, the model might inherit and amplify biases present in the Lizard dataset. This issue is mentioned but not explored experimentally.

References:
[1] Cechnicka, S., Ball, J., Baugh, M., Reynaud, H., Simmonds, N., Smith, A. P., Horsfield, C., Roufosse, C., & Kainz, B. (2024). URCDM: Ultra-Resolution Image Synthesis in Histopathology. In International Conference on Medical Image Computing and Computer-Assisted Intervention (pp. 535–545). Springer Nature Switzerland.

[2] Harb, R., Pock, T., & Müller, H. (2024). Diffusion-based generation of Histopathological Whole Slide Images at a Gigapixel scale. In Proceedings of the IEEE/CVF Winter Conference on Applications of Computer Vision (WACV) (pp. 5119–5128). IEEE. https://doi.org/10.1109/WACV57701.2024.00505

[3] Min, S., Oh, H.-J., & Jeong, W.-K. (2024). Co-synthesis of Histopathology Nuclei Image-Label Pairs using a Context-Conditioned Joint Diffusion Model. arXiv preprint arXiv:2407.14434. https://arxiv.org/abs/2407.14434

[4] Pozzi, M., Noei, S., Robbi, E., Cima, L., Moroni, M., Munari, E., Torresani, E., & Jurman, G. (2024). Generating and evaluating synthetic data in digital pathology through diffusion models. Scientific Reports, 14, 28435. https://doi.org/10.1038/s41598-024-79602-w

[5] Cechnicka S, Ball J, Reynaud H, Arthurs C, Roufosse C, Kainz B. Realistic data enrichment for robust image segmentation in histopathology. InMICCAI Workshop on Domain Adaptation and Representation Transfer 2023 Oct 12 (pp. 63-72). Cham: Springer Nature Switzerland.

[6] Wang, T., Song, Y., Cao, J., Song, J., Wang, S., Wang, C., & Zhu, J. (2022). Semantic Image Synthesis with Diffusion Models. In Proceedings of the IEEE/CVF Conference on Computer Vision and Pattern Recognition (CVPR) (pp. 14454–14465). IEEE.

---

### Review · Reviewer_vVw2 · 2025-04-28

**Summary Of Contributions:**

The authors introduce a novel nuclei-aware semantic diffusion framework for generating realistic H&E-stained histology image patches that are conditionally annotated with multiple nuclei types. The core idea is a two-stage diffusion-based pipeline: first a diffusion model generates a multi-class nuclei mask given a desired combination of six cell types (epithelial, lymphocyte, connective tissue, neutrophil, plasma, eosinophil)​, and then a second U-Net-based diffusion model synthesizes an H&E image patch conditioned on that mask.

- Algorithm: The mask generator uses a multi-hot vector of selected nuclei types injected into the U-Net’s time embeddings (enabling pixel-perfect localization of each nucleus), and the image generator incorporates the full semantic map (including an “edge” channel) via spatially-adaptive normalization layers​

- Training: Using the publicly available Lizard dataset, the proposed framework demonstrates superior performance in generating realistic synthetic tissue patches. Quantitative metrics such as the Fréchet Inception Distance (FID) and Inception Score (IS) show that their method outperforms existing GAN-based approaches and a recent morphology-focused diffusion model in generating samples that closely resemble real data. For instance, their NASDM (Real Masks) achieved an FID of 14.1 on the colon dataset, compared to 18.8 for Morph-Diffusion.

- Demonstrated Efficacy for Downstream Tasks: The experiments show that synthetic patches generated by the framework can effectively augment real datasets to improve segmentation model performance (Table 3). Furthermore, models trained solely on synthetic data achieve comparable performance to those trained on real data (Table 4), suggesting that the synthetic data can potentially serve as a viable alternative or supplement to real, annotated data for training. An evaluation of utility for downstream tasks, showing that synthetic data can effectively augment nuclei segmentation training, with segmentation accuracy improving as more synthetic patches are added (Dice up to ~0.809, Table 3​) and remaining stable when replacing real with synthetic data.

- Qualitative Validation by Expert Pathologists: The generated patches were evaluated by 2 board-certified pathologists, providing crucial qualitative feedback on their realism, the accuracy of nuclei representation, and consistency with the conditioning masks. The expert evaluation indicated comparable realism between patches generated from real and synthetic masks (Figure 6).

Overall, the paper presents a comprehensive technical advance in semantic image synthesis for computational pathology, with strong empirical results and evaluation.

**Audience:**

Yes

**Broader Impact Concerns:**

- Bias and generalization. The authors acknowledge that the model reproduces biases in the training data​. Indeed, a diffusion model cannot generate cell types or patterns it hasn’t seen. This could inadvertently bias downstream algorithms: for example, rare tumor morphologies or staining variations not in the Lizard dataset would never be synthesized, potentially limiting the diversity of augmented data. We recommend the authors emphasize this risk: any clinical deployment must ensure that synthetic augmentation does not omit clinically important variants.

- Clinical validity of synthetic data. While synthetic images help with data scarcity, there is a risk if such images were mistaken for real patient data. The paper should clarify that synthetic patches are meant for training/benchmarking models (not for diagnosis). Proper disclaimers should be used if sharing synthetic datasets. Moreover, model users should be aware that despite pathologist validation, the images lack full biological context; e.g. subtle tissue architecture cues outside the patch are absent​.

- The paper demonstrates promising potential for practical medical applications. Achieves state-of-the-art generative metrics (FID: 14.1) and integrates semantic conditioning, ensuring precise nuclei localization (validated by pathologists). Downstream segmentation models trained on synthetic data match real-data performance (Table 4), proving utility in real-world tasks.

**Claims And Evidence:**

Yes

**Requested Changes:**

- Evaluate on additional data or tasks. At minimum, demonstrate generation on another histopathology dataset or at least a held-out subset with different tissue (if available). For example, a small experiment on a different organ or a public histology image set would address concerns about overfitting to colon tissue​. If new data is not available, a convincing analysis (or citation of related work) about the generality of the conditioning scheme is needed.

- Clarify and strengthen baselines. Provide a clearer comparison with non-conditional generative baselines. For instance, train an unconditional diffusion or GAN on the same patches and report its FID/IS on the colon data, to confirm that conditioning indeed gives an edge. If infeasible, cite results or include a short discussion illustrating why those methods fail at instance-level control. Also, ensure Table 1’s comparisons are fair: if possible, rerun Morph-Diffusion (Moghadam et al., 2023) with colon data or clarify why the reported values are from different domains. In my opinion, that can provide strong baselines to compare.

**Strengths And Weaknesses:**

Strength:

- Novel and well-motivated approach. The idea of using conditional diffusion to generate annotated histology images is original. By conditioning on an exact nuclei mask, the model guarantees pixel-level control over cell locations and types, which is valuable for creating labeled training data. Prior work mainly used GANs or unconditioned models; here the semantic diffusion approach is a clear innovation. The paper clearly explains the methodological design (Sec.3, Fig.1) and positions it well with respect to recent diffusion (Ho et al., 2020​; Dhariwal & Nichol 2021) and semantic diffusion literature (Wang et al., 2022​).

- Comprehensive experiments. The authors conduct a rich set of evaluations. Quantitatively, they report FID and IS scores showing that NASDM outperforms existing approaches by a significant margin (e.g. FID=14.1 vs 18.8 for closest competitor​). They also include ablations on model variants (e.g. NASDM vs NASDM++ with generated masks) and technical choices (dropout guidance, SiLU vs ReLU, etc. – although details of these ablations could be more clearly highlighted). Qualitatively, figures of synthetic patches illustrate varied cell arrangements that look realistic. Crucially, the downstream segmentation tests (Sec.4.3) show a clear benefit: adding synthetic patches to the HoVerNet training improves Dice scores over the baseline (Table 3​), and replacing real data entirely with synthetic yields comparable performance. These experiments convincingly support the claim that the generated data is useful.

- Expert validation. Including a user study with board-certified 2 pathologists is a major strength. The pathologists found no statistically significant differences between synthetic and real patches across four key criteria​.
. This not only bolsters confidence in the visual fidelity of the outputs, but also addresses a critical question in medical imaging: whether synthetic images “look real” to domain experts. Many papers lack such human studies, so this is a notable plus.

- Clarity and presentation. The paper is generally well-written and organized. Equations (1–7) are clearly presented, and the architecture details (multi-hot injection, spatial normalization in decoder) are explained with sufficient clarity​.

- Technical thoroughness. The implementation details (optimizer, learning rates, guidance dropout, number of diffusion steps) are well specified​. The use of a public dataset and plan to release code enhances reproducibility. The study explores both “real mask” and “synthetic mask” modes (NASDM vs NASDM++) to show robustness.


Weakness:

- Limited dataset and domain. All experiments are on the Lizard colon histology dataset (238 images of colon tissue). It is unclear how well the approach would generalize to other tissue types or staining protocols. The authors acknowledge this limitation (“model was specifically demonstrated on colonic tissue”, uncertain if it will generalize​). Testing on at least one additional dataset (e.g. a different organ or cancer type) would strengthen the claim of general applicability. As it stands, the method may be over-fit to colon histology appearances and annotations.

- Evaluation scope. Relatedly, the evaluation focuses on small $128\times 128$ patches. While this is a sensible starting point, real clinical histology evaluation is usually on whole-slide images at multiple scales. The user study and segmentation experiments use these patches, so it is uncertain how the model performs on larger or heterogeneous tissue contexts. The authors note this (evaluation on patches may not capture broader architecture​). Additional experiments on larger fields-of-view (or tiling patches into larger synthetic tissue) would be valuable.

- Quantitative metrics and statistical analysis. The segmentation results in Tables 3–5 show small improvements (e.g. Dice rising from 0.7713 to 0.8092 in Table 3​). While improvements appear consistent, it would be good to report statistical significance or effect sizes more explicitly. Are these improvements practically significant for pathology? Similarly, the FID/IS improvements are substantial, but since FID can vary with implementation, some discussion of variance (or reporting means over multiple runs) would strengthen confidence. The authors do report standard deviations for segmentation (±0.0004, etc.), which is good, but similar stats for generation metrics (FID) are missing.

- Minor clarity issues. A few notational typos (e.g. "equation equation 6" in Sec.4.1​) should be fixed. Also, the term "NASDM++" for generated masks is introduced somewhat abruptly in Table 1 caption; a brief explanation in the text would help. Overall, however, these are minor editorial points.

---

### Review · Reviewer_NMCJ · 2025-05-09

**Summary Of Contributions:**

The paper presents conditioned diffusion models for histopathology. The authors present a method that (1) generates cell segmentation masks conditioned on a one-hot vector of (multiple) cell types, and (2) generates a patch conditioned on a cell segmentation mask (either real or synthetic produced with method (1)). The method is based on previous work, but applied to histopathology. The methods are extensively evaluated on the generative tasks itself and to use the synthetic data for downstream cell segmentation tasks with varying real-synthetic dataset setups.

Contributions and new knowledge: A downstream segmentation model trained on only synthetic patches generated from real(!) segmentation masks using a conditioned diffusion model achieves similar performance to a segmentation model trained on the same real segmentation masks and their respective real patches.

**Audience:**

Yes

**Broader Impact Concerns:**

There are no concerns on ethical implementations of the work.

**Claims And Evidence:**

Yes

**Requested Changes:**

The reviewer suggests to clearly define all mathematical symbols used, and more clearly describe the method, its novelties and design choices. For example:
- "a noise prediction-based formulation results in superior image quality"; how is this determined? Where are these results presented?
- $\epsilon$ and $\theta$ in eq (3) is not defined
- "[...] the given simplified loss function does not provide a training signal for $\sum_\theta(x_t,y,t)$. What is this part of the loss? Why is the loss function simplified?
- What is $L_{\text{vlb}}$ ?
- The reviewer finds it unclear why the addition of the nuclei edge map is beneficial, or why this design choice was made

Explicitly stating that the synthetic images for downstream task training are generated from real annotations in Figure 4 may make the data generation setup more clear to the reader. E.g. "synthetic" may be replaced with "semi-synthetic", while "synthetic" is reserved for data that is generated using (1) synthetically generated one-hot vector, which are (2) used for synthetic segmentation mask generation, which are used for (3) synthetic images.

The reviewer believes it would generate more knowledge with practical impact if the authors perform downstream task experiments with synthetic images generated from synthetic masks generated from synthetic one-hot vectors of cell types in a setting that resembles the situation where there are few annotations for e.g. rare cancer types

A more detailed discussion of the qualitative experiments could provide more information. Detailed bar plots are shown, but the results section states "[...] perform comparably across all four criteria", which appears to be a very quick conclusion that does not generate much knowledge.

Although the authors present the limitations and future work, the discussion of the results is foregone. Adding a discussion that critically interprets the results and places them in perspective of the problem presented in the introduction (limited annotations for rare cancer types) would be beneficial.

**Strengths And Weaknesses:**

Strengths

The work presents a possible solution for an important problem: Dealing with limited labelled data in the domain of computational pathology.
The potential benefit of training on synthetically generated images is reasonably quantitatively evaluated.
Additionally, the quality of the generated images are qualitatively validated by board-certified pathologists.

Weaknesses
The introduction focuses particularly on rare cancer types, but the experimental setup and discussion does not state anything about this. Experiments are performed on a colorectal cancer set. All quantitative results of the downstream task (nucleus segmentation) utilize synthetic images generated from real segmentation masks. The practical value lies in completely synthetic images due to the lack of such detailed segmentation masks, yet these are only qualitatively evaluated.

Authors state that they design a novel first-of-its-kind method, but the novelties are unclear compared to the approach in "Semantic Image Synthesis via Diffusion Models" to the reviewer.

The reviewer finds it difficult to follow the description of the proposed method. Many mathematical symbols are not defined. Similarly, the reviewer found it difficult during a first read to distinguish between the method that generates segmentation masks from the one-hot vector and the synthetic images from a (real or synthetic) segmentation mask. In the same line, the experiments utilize many different real/synthetic setups that the reviewer found difficult to track. E.g., the quantitative results are only performed on synthetic images generated from real segmentation masks (of which the practical value seems limited), while the qualitative evaluation is performed for synthetic images generated from synthetic segmentation masks. In the same line, the method's name in Table 1 (NASDM) is not defined?

The experiment summarized in Table 3 has varying dataset sizes, yet it appears the authors train for 100 epochs for each augmented dataset. The difference in results may be due to the increase in training steps.

It is unclear to the reviewer if a validation set was used for model selection during training for the downstream task. E.g. "The models are validated after every two epochs during training, and we report the metrics of the best-performing model on the test set".

---

### Decision · Action_Editor_JL9m · 2025-09-01

**Recommendation:** Reject

**Audience:**

No

**Audience Explanation:**

Yes, there exist researchers that would be interested in such a line of work. However, it is likely that the paper would be a better fit to a journal that focuses on medical AI.

**Claims And Evidence:**

No

**Claims Explanation:**

The reviewers have doubts on the experimental results and generality of the claims. Two out of three reviewers note that the introduction refers to rare cancers but experiments not, and experiments are only done on the Lizard colon histology dataset.

Further, there are strong doubts on all other major aspects of evaluating a paper, including positioning, and description of method.

**Resubmission Of Major Revision:**

The authors may consider submitting a major revision at a later time.